# Visual feature analysis on selective appetite in individuals with autism spectrum disorders

**Kazunori Terada[1], Taku Imaizumi[2], Kazuhiro Ueda[2], Natsuki Nishikawa[3], Haruto Yoshida[1], Yukiya Taki[1], Shunsuke Fujii[1], Lu Li[2], Masashi Komori[4], Kunihito Kato[1], Hirokazu Kumazaki[3]***

**1** Faculty of Engineering, Department of Electrical, Electronic, and Computer Engineering, Gifu University, Gifu, Japan, **2** Graduate School of Interdisciplinary Information Studies, The University of Tokyo, Tokyo, Japan, **3** Department of Neuropsychiatry, Graduate School of Biomedical Sciences, Nagasaki University, Nagasaki, Japan, **4** Handling of Personal Information at Osaka Electro-Communication University, Osaka, Japan

* kumazaki@tiara.ocn.ne.jp

## Abstract

### Background

Individuals with autism spectrum disorders (ASD) experience more severe selective eating problems than their neurotypical peers. Identifying the causes of selective eating behavior poses a considerable challenge, even for caregivers. Accurate identification of the underlying causes of this behavior is essential for developing interventions aimed at overcoming dysfunctional, unbalanced diets. However, studies that meticulously identify the causes of selective eating behaviors are scarce. This investigation aims to explore the differences in preferences for sunny-side-up eggs between individuals with ASD and those with typical development (TD), focusing on the factors influencing their likes and dislikes through a systematic analysis of visual features.

### Method

Thirty-nine individuals with ASD (mean age, 23.4±4.7 years; 82% men) and fifty individuals with TD (mean age, 22.2±1.3 years; 64% men) participated in this study. We used a total of 50 images of sunny-side-up eggs as visual stimuli. Using Non-negative Matrix Factorization and Decision Tree analysis, factors associated with visual preferences for sunny-side-up eggs were identified.

### Data and Results

We could identify factors associated with visual preferences for sunny-side-up eggs. Subsequent linear regression analysis provided insight into how these visual features delineate preference boundaries between liked and disliked foods, with noteworthy distinctions emerging between the ASD and TD groups.

**Data availability statement:** All relevant data are within the paper and its Supporting Information files.

**Funding:** This work was partially supported by a Grants-in-Aid for Scientific Research from the Japan Society for the Promotion of Science (23H04358; 24H01554). The funders had no role in study design, data collection and analysis, decision to publish, or preparation of the manuscript.

**Competing interests:** The authors report no conflicts of interest.

## Conclusions

This study provides novel insights into the visual determinants of food preferences in individuals with ASD through systematic analysis of image features. Our findings indicated the potential to predict preferences while elucidating the causes of selective eating behaviors, thereby offering solutions for individuals with ASD.

## Introduction

Repetitive and restricted behaviors (RRBs) are common symptoms of autism spectrum disorders (ASD) [1]. RRBs encompass intense, fixated interest in a specific activity, object, or subject [2]. Research has indicated that individuals with ASD encounter more severe selective eating problems compared to their neurotypical peers [3–6]. These behaviors can lead to nutritional imbalances, such as excessive intake of junk food, contributing to weight gain, disease predisposition, and stunted physical growth and ultimately impacting long-term outcomes if left unaddressed [7–9]. Consequently, caregivers and clinicians are tasked with managing the daily challenges of various dysfunctional meal-time behaviors, including food refusal, limited food variety, reliance on single food types, or exclusive liquid diets [10].

Determining the causes of selective eating behaviors is difficult even for caregivers. Accurately identifying these causes is crucial for establishing programs to address the dysfunctional and unbalanced diets of individuals with ASD. The causes of selective eating behaviors are thought to depend on various medical, sensory, or behavioral factors [11]. Previous studies have indicated that sensory sensitivity and anxiety are associated with selective eating behavior [12,13]. Individuals explore food through sensory modalities, gradually acquiring knowledge through vision, taste, tactile, and smell [14,15]. Notably, visual elements such as color and shape, along with color-shape associations, play an essential role in shaping food preferences among individuals with ASD [14]. Nevertheless, studies that detail the causes of selected food behaviors from a visual perspective remain limited.

A statistical method to extract common features from various images [16] reportedly can evaluate the composition and quality of food products [17] and even predict the freshness of spinach [18]. Through visual feature extraction, it is possible to identify characteristics common to images.

This study aims to investigate the differences in preference for, and the factors underlying, the liking and disliking of sunny-side-up eggs among individuals with ASD compared to typical development (TD) individuals through a systematic analysis of image features. Visual features were calculated for sunny-side-up eggs, and the characteristics typical of images preferred or avoided by individuals with ASD were extracted. The choice of sunny-side-up eggs was driven by several factors: 1) they represent a straightforward yet well-defined stimulus for investigating visual preferences, with clearly delineated elements (such as the white and yolk) that allow for measurable variations in color, shape, and spatial relationships; 2) our preliminary study suggests diverse reasons for the liking or disliking of their shape; and 3)

sunny-side-up eggs constitute one of the most commonly consumed dishes worldwide. It is hypothesized that the results of this study will highlight a primary distinction in liking and disliking of foods and elucidate the causes of feeding issues faced by individuals with ASD.

## Materials and methods

### Participants

We conducted an a priori power analysis using G*Power within a linear regression framework, assuming a medium effect size ($f^2 = 0.15$), an alpha level of .05, and a power of .80. This analysis indicated that approximately 77 participants (i.e., about 39 participants per group) would be required. Considering potential missing data or insincere response into account, we aimed to recruit 50 participants per group. Participants were recruited through flyers detailing the content of the experiment. The inclusion criteria for individuals with ASD were as follows: 1) those aged between 18 and 30 years, 2) those diagnosed with ASD based on the Diagnostic and Statistical Manual of Mental Disorders, Fifth Edition (DSM-5), by a supervising study psychiatrist, and 3) those not currently taking medication. All of them had no regular jobs. Participants were recruited between 03/03/2022, and 10/23/2023. At enrollment, the diagnoses of all participants were confirmed by a psychiatrist with more than fifteen years of experience in ASD using standardized criteria from the Diagnostic Interview for Social and Communication Disorders (DISCO) [19], which possesses robust psychometric properties [20]. The focus on young adults in this study is justified by their sufficient vocabulary to articulate preferences and respond to questionnaires, as well as the importance of eating habits acquired at this age for weight gain and disease susceptibility.

Conversely, individuals with TD were recruited from a public offering. The inclusion criteria for the TD group were as follows: 1) those aged between 18 and 30 years, 2) those attending mainstream school, 3) those with no evidence of intellectual impairment and developmental disorders, and 4) those diagnosed as TD by a qualified psychiatrist based on developmental and lifestyle information, including Autism Quotient (AQ). In total, 39 individuals with ASD (mean age = 23.4 ± 4.7, 82% men) and 50 individuals with TD (mean age = 22.2 ± 1.3, 64% men) participated in this study.

### Ethical procedure

The present study received approval from the Ethics Committee of Gifu University and was conducted in accordance with the Declaration of Helsinki. Following a comprehensive explanation of the study, all participants and their guardians provided informed consent for participation. A separate written informed consent was obtained from participants and/or legal guardians (for minors) for the publication of any potentially identifiable images or data included in this research.

### Self-administered questionnaire

Participants completed the Autism Spectrum Quotient-Japanese version (AQ-J) [21], a self-administered questionnaire designed to measure autistic traits and evaluate ASD-specific behaviors and symptoms. The AQ-J comprises five subscales (social skills, attention switching, attention to detail, imagination, and communication). Prior work with the AQ-J has established its reliability across cultures [22] and ages [23]. The AQ is sensitive to the broader autism phenotype [24]. In this study, the AQ-J score was not utilized as a cutoff for ASD; rather, the DSM-5 and Diagnostic Interview for Social and Communication Disorders (DISCO) were employed for diagnosis and participant inclusion. The AQ-J score served solely as a factor for ruling out ASD in the TD group.

Full-scale IQ scores were obtained using the Japanese Adult Reading Test (JART), a standardized cognitive assessment tool utilized to estimate premorbid intelligence quotient (IQ) of individuals with cognitive impairments [25]. The JART demonstrates good validity in measuring IQ and yields results comparable to those of the Wechsler Adult Intelligence Scale – third edition (WAIS-III) [25].

The Adolescent/Adult Sensory Profile (AASP) is a self-administered questionnaire measuring sensory processing in individuals aged ≥11 years [26]. The internal consistency coefficients of the AASP range from 0.64 to 0.78 for the quadrant scores. Before the experiment, participants indicated how frequently they exhibited specific behaviors related to sensory experiences using a scale from one (almost never) to five (almost always). The AASP encompasses four quadrants of sensory processing: low registration, sensation seeking, sensory sensitivity, and sensation avoiding. As the AASP does not categorize responses according to perceptual domains (e.g., auditory, visual, and tactile), a perceptual domain analysis was not performed in this study.

## Visual stimulus

We employed a total of 50 images of sunny-side-up eggs as visual stimuli. Most of these images [47 of 50] were photographs of real sunny-side-up eggs, supplemented with two images of imitation food samples and one computer-generated graphic (CG). These real egg images were selected to encompass a wide range of visual presentations. Initially, 200 sunny-side-up eggs were cooked and photographed under controlled conditions. Standardized lighting, camera settings, and positioning were maintained consistently across all images (see Fig 1), ensuring that variations in stimuli were attributed to individual differences in the eggs rather than photographic conditions. From this initial compilation, 47 images representing a diverse spectrum of visual attributes were selected, ranging from typically attractive, well-formed sunny-side-up eggs to less appealing, imperfectly cooked examples.

The inclusion of two imitation food samples and one CG image was intended to broaden the range of visual representations beyond what could be achieved with real egg photographs alone. The CG image was specifically designed to depict a sunny-side-up egg characterized by a perfectly smooth, glossy surface and an ideal shape. This inclusion allowed for an examination of how various types of visual representations, including idealized or artificial versions, might influence participants' responses. The imitation food samples, crafted by professional food modelers, showcased idealized sunny-side-up eggs commonly utilized in food advertising. All images were standardized to a JPG format with a resolution of 320 × 320 pixels and a color depth of 24-bit.

## Procedure

The experiment was conducted in a laboratory environment using computers operating on the Qualtrics XM Platform (Version 2023.09) via Google Chrome browsers. Participants were situated in individual, enclosed, sound-attenuated

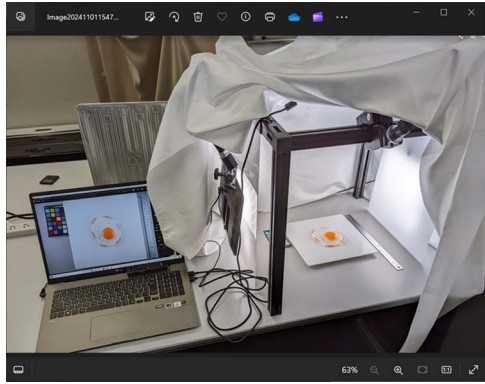

**Fig 1. Photography setup for sunny-side-up egg images.** The setup consists of a light box with LED illumination and a calibrated camera system connected to a laptop computer. A white fabric draped over the entire setup blocks external light to ensure controlled lighting conditions. A color checker chart is included in each image for color calibration. All images were captured under these standardized conditions to ensure consistent illumination across samples.

spaces, each equipped with a desk and PC. Upon arrival, participants received a briefing on the experiment and provided informed consent through the Qualtrics interface. Following this, they completed a demographic questionnaire, which took approximately 5 min.

The primary experimental task entailed the viewing and rating of 50 images of sunny-side-up eggs, presented in a random order to mitigate potential order effects. Each image was displayed at approximately the center of the participants' browser window, with no time limit imposed for viewing individual images. For each image, participants were asked to rate their desire to eat the depicted egg using a visual analog scale (VAS). Participants rated their desire on a 101-point scale ranging from 0 (do not want to eat at all), through 50 (neither), to 100 (very much want to eat). Simultaneously, participants were invited to provide optional free-text explanations for their reasons for wanting or not wanting to consume the egg. Responses, including VAS ratings and free-text explanations, were recorded in real-time via Qualtrics and stored on the server. The complete experimental procedure lasted an average of approximately 70 min. The primary quantitative measure in this study was the rating of desire to eat each of the 50 sunny-side-up egg images.

### Analysis of demographic data

Statistical analyses were conducted using SPSS software (version 27.0; IBM, Armonk, NY, USA). Descriptive statistics were computed for both participant groups (i.e., individuals with ASD and those with TD). The full-scale IQ scores, AQ-J scores, and AASP sub-scores were compared between the ASD and TD groups using a $t$ test. Age comparisons between participants were performed using the Mann–Whitney U test. The differences in sex ratios were analyzed using $\chi^2$ tests and $t$ tests.

### Analysis for elucidating differences in visual preferences for sunny-side-up egg images between individuals with ASD and TD individuals

The analysis approach comprised four complementary steps designed to comprehensively characterize differences in visual preferences for sunny-side-up egg images among individuals with ASD and those with TD. First, images were statistically classified as "liked" or "disliked" based on participant ratings to establish baseline preference patterns in each group. Second, underlying visual features were extracted from the images using Non-negative Matrix Factorization (NMF), selected for its capacity to identify interpretable, part-based visual components. Third, decision tree analysis was utilized to determine which combinations of these visual features were most predictive of food preferences, providing a quantitative measure of feature importance. Finally, linear regression analysis was conducted using the identified key features to visualize and compare preference boundaries between groups. This systematic combination of analyses facilitated identification of the visual features influencing food preferences while allowing for a quantitative characterization of how these influences differ between individuals with ASD and TD individuals, providing insights into the visual criteria underlying selective eating behavior in each group.

### Image preference classification

To categorize sunny-side-up egg images based on participants' preferences, the following methodology was employed. First, mean liking scores were calculated for each image separately for the ASD and TD groups. Subsequently, each mean score was compared to the neutral score of 50 using one-sample $t$ tests. Images were then classified as follows:

- "Liking": Mean score significantly >50 ($p < 0.05$)

- "Disliking": Mean score significantly <50 ($p < 0.05$)

- "Neutral": No significant difference from 50

This classification was executed independently for the ASD and TD groups, enabling comparison of preference patterns across groups and visualization of these patterns. The distribution difference among the groups (i.e., liking, disliking, and neutral) was analyzed using χ2 tests and *t* tests.

### Non-negative matrix factorization analysis

To identify key visual features that could influence preferences, NMF was employed [27]. NMF was selected owing to its ability to generate part-based representations of images [28], aligning with research suggesting that some individuals with ASD may demonstrate enhanced abilities in processing local features, although the extent and specificity of this trait remain subjects of ongoing inquiry [29,30]. The part-based representations derived from NMF may prove especially useful in investigating these potential processing differences. The number of NMF components was determined based on preliminary analyses utilizing Principal Component Analysis (PCA) [31], given the absence of a standardized criterion for determining the dimensionality of base vectors in NMF. As an additional validation step, the explanatory power of NMF and PCA was compared.

### Decision tree analysis

To ascertain which extracted visual features most strongly influence judgments, a decision tree analysis was conducted. The features extracted via NMF served as input variables, whereas the image preference classifications served as the output variable. The analysis utilized the Classification and Regression Trees algorithm, as implemented in scikit-learn (version 1.4.2). Feature importance scores were calculated based on the decrease in Gini impurity.

### Linear regression analysis

We conducted a linear regression analysis to examine potential differences in the discriminant boundary between the ASD and TD groups. This analysis concentrated on the most significant features identified in the decision tree analysis. Scatter plots were created for the sunny-side-up egg images using these NMF dimensions, with feature importance scores serving as independent variables and preference ratings as the dependent variable. Regression analyses were conducted independently for individuals with ASD and TD individuals to compare discriminant boundaries.

## Results

### Demographic data

All participants successfully completed the experimental procedures and questionnaires. No significant differences were observed in age, low registration scores, sensation-seeking scores, or sensation-avoiding scores between individuals with ASD and TD individuals. In contrast, there were substantial differences in the IQ, AQ-J scores, and sensory sensitivity scores between the two groups. Detailed results are presented in Table 1.

### Image preference classification

The results of the classification of sunny-side-up egg images based on the questionnaire responses are illustrated in Fig 2. The distribution of preferences (liking, disliking, and neutral) was 8:20:22 for individuals with ASD and 20:19:11 for individuals with TD, with a significant difference identified between groups ($\chi^2(2, N = 100) = 8.84, p = .012$). The classification results categorized the sunny-side-up egg images into five distinct labels: (1) images liked by both groups (n = 8), (2) images liked solely by individuals with TD (n = 12), (3) images disliked by both groups (n = 19), (4) images disliked exclusively by the individuals with ASD (n = 1), and (5) images that did not belong to either group (n = 10). Notably, no images were liked exclusively by the ASD group, nor were there any images disliked only by the TD group. The eight images (Fig 1, top row) liked by the ASD group were also favored by the TD group. All 19 images (Fig 1, third row from the

**Table 1. Descriptive statistics of participants.**

| | ASD (n = 39) Mean (Standard error of mean) | TD (n = 50) Mean (Standard error of mean) | Statistics | | |
|---|---|---|---|---|---|
| | | | U or t or χ² | df | p |
| Age | 22.92 (0.58) | 22.16 (0.19) | U = 946.00 | | 0.808 |
| Sex (Male: Female) | 32:7 | 32:18 | χ² = 3.534 | 1 | 0.060 |
| Full-scale IQ | 99.21 (1.49) | 110.48 (0.70) | t = −7.339 | 87 | 0.000** |
| AQ-J | 27.03 (0.94) | 20.82 (1.12) | t = 4.102 | 87 | 0.000** |
| AASP | | | | | |
| Low Registration | 38.33 (1.67) | 35.94 (0.94) | t = 1.320 | 87 | 0.190 |
| Sensation Seeking | 37.71 (1.11) | 41.18 (1.14) | t = −2.138 | 87 | 0.035* |
| Sensory Sensitivity | 36.64 (1.70) | 37.22 (1.14) | t = −0.397 | 87 | 0.692 |
| Sensation Avoiding | 37.87 (1.82) | 37.70 (1.12) | t = 0.084 | 87 | 0.933 |

\** $p < 0.01$, \* $p < 0.05$.

ASD, autism spectrum disorder; TD, typical development; AQ-J, Autism Spectrum Quotient, Japanese version. Higher scores on the AQ-J reflect a greater number of ASD-specific behaviors; AASP, Adolescent/Adult Sensory Profile; IQ, intelligence quotient; U, Mann–Whitney U test; t, t test; χ², chi-squared text; df, degrees of freedom.

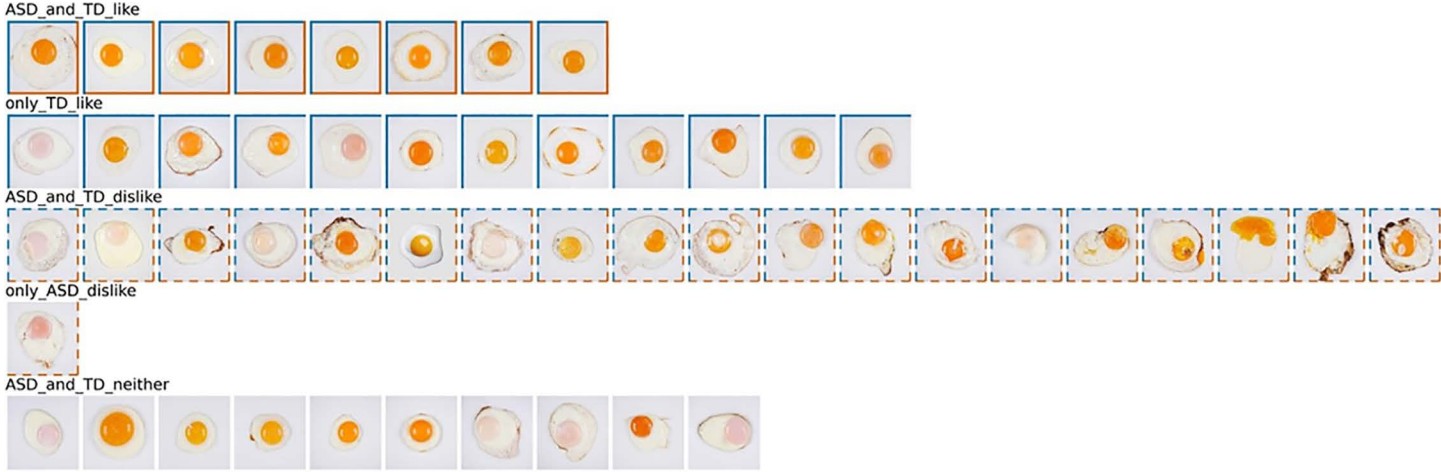

**Fig 2. Classification of sunny-side-up egg images based on the questionnaire.** The results showed a difference in the distribution of visual preferences between individuals with ASD and TD. In the top row, there are 8 images that both the ASD and TD groups liked, 12 images that only the TD group liked, 19 images that both the ASD and TD groups disliked, 1 image that only the ASD group disliked, and 10 images that did not belong to either of the two groups. All images are ordered according to the mean preference scores of the ASD group. ASD, autism spectrum disorder; TD, typical development.

top) disliked by the TD group were similarly disliked by individuals with ASD. Among the three artificial samples, one food sample (Fig 2, top row, sixth from the left) was classified as "liked by both ASD and TD groups," another food sample (Fig 2, second row from the top, eighth from the left) was categorized as "liked only by the TD group," and the CG image (Fig 2, third row from the top, sixth from the left) was classified as "disliked by both groups.".

## NMF analysis

The NMF analysis facilitated the extraction of key visual features from the sunny-side-up egg images. A total of 11 base vectors were extracted, following the standard criteria for dimensional selection (refer to S1 Appendix and S1 Table in

Supplementary Material for details regarding dimension determination). Fig 3 provides a visualization of these NMF base vectors. Factor 1 exhibited highly prototypical images (i.e., a clean, double-circle shape and absence of burn marks). Comparative analyses confirmed that NMF yielded greater explanatory power than PCA in predicting preference ratings across both groups (see S2 Appendix and S2 Table in Supplementary Material for detailed analyses).

## Decision tree analysis

Decision tree analysis employed the 11 NMF dimensions as independent variables and the five image preference categories as dependent variables. The analysis was conducted to a maximum tree depth of four, continuing until the decision tree accuracy exceeded 80%. The importance of each feature dimension was calculated. Factor 1 emerged as the most essential feature (0.469), followed by Factor 6 and Factor 8 (0.160 and 0.099, respectively). The importance of NMF in each factor is summarized in Table 2. Importantly, the order of NMF factors indicates the relevance in characterizing the objective image structure, whereas the importance scores reflect the predictive efficacy of each factor concerning subjective preferences. These two aspects may not necessarily align, as features that effectively describe image data may differ from those that influence human preferences.

## Linear regression analysis

Scatter plots illustrating Factor 1 versus Factor 6 (Fig 4) and Factor 1 versus Factor 8 (Fig 5) were created to depict the discriminant boundaries for the ASD and TD groups based on linear regression analysis. The discriminant boundary for the ASD group consistently appeared to the right of that for the TD group in both analyses, although this difference did not reach statistical significance Table 3.

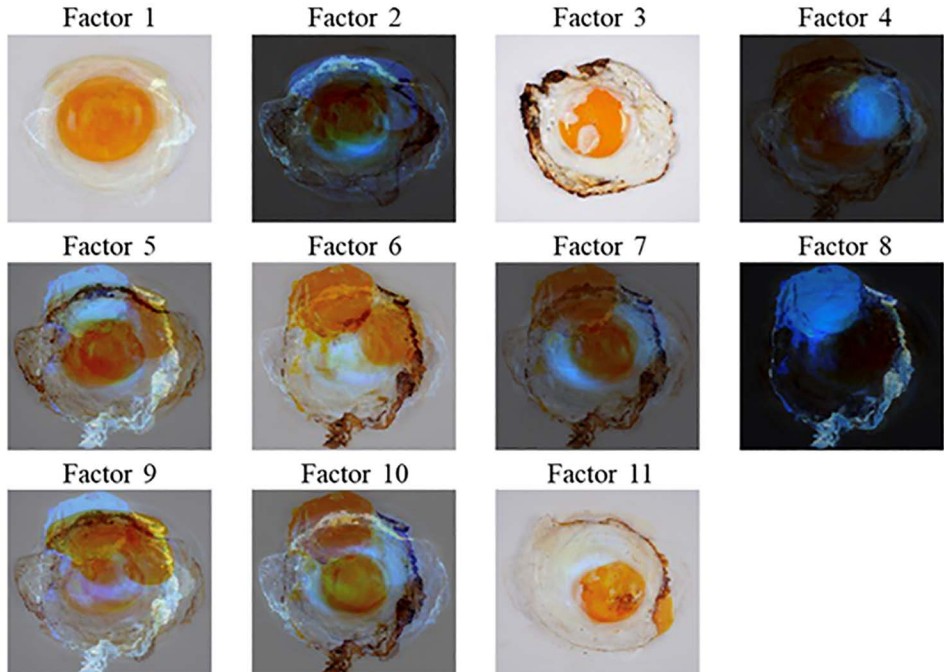

**Fig 3. Visualization of base vectors extracted by NMF.** NMF, non-negative matrix factorization.

 

**Table 2. Importance of NMF in each factor.**

| Dimension | Importance |
|---|---|
| Factor 1 | 0.469 |
| Factor 2 | 0.000 |
| Factor 3 | 0.050 |
| Factor 4 | 0.036 |
| Factor 5 | 0.073 |
| Factor 6 | 0.160 |
| Factor 7 | 0.072 |
| Factor 8 | 0.099 |
| Factor 9 | 0.000 |
| Factor 10 | 0.000 |
| Factor 11 | 0.042 |

NMF: non-negative matrix factorization.

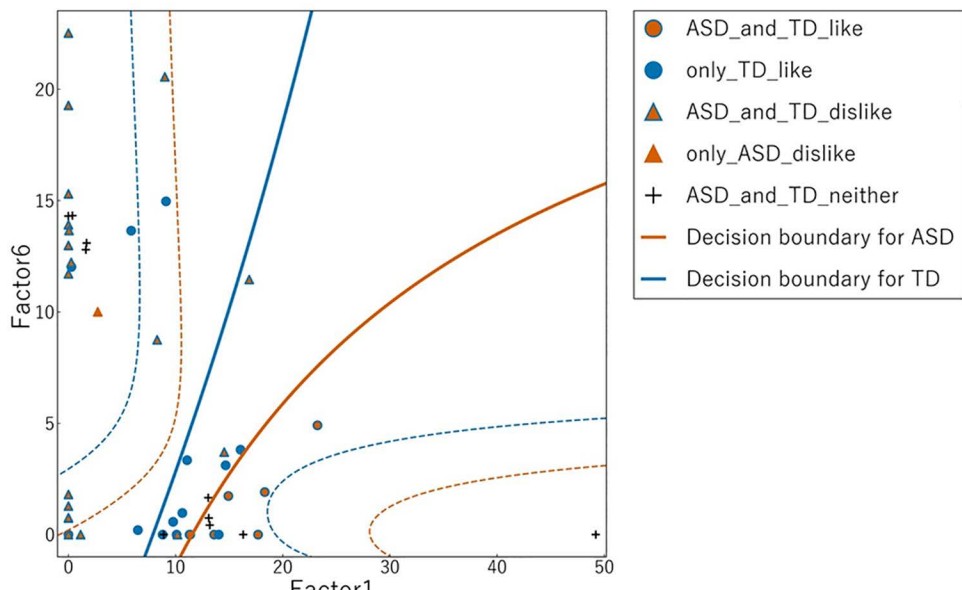

**Fig 4. Scatter plot of sunny-side-up egg images based on NMF Factor 1 and Factor 6.** The horizontal axis represents Factor 1 and the vertical axis represents Factor 6. Blue markers represent ratings by the ASD group, and orange markers represent ratings by the TD group. Circles indicate images rated as "Liking," whereas triangles indicate images rated as "Disliking." Solid lines show the linear regression boundaries between "Liking" and "Disliking" ratings for each group (blue for ASD and orange for TD). Dotted lines represent the 95% confidence intervals for these boundaries. ASD, autism spectrum disorder; TD, typical development; NMF, non-negative matrix factorization.

## Discussion

The primary objective of this study was to examine the differences and underlying causes of visual preferences for sunny-side-up egg images between individuals with ASD and those with TD. By utilizing NMF and decision tree analysis, we could identify factors associated with visual preferences for sunny-side-up eggs. Subsequent linear regression analysis provided insight into how these visual features delineate preference boundaries between liked and disliked foods, with

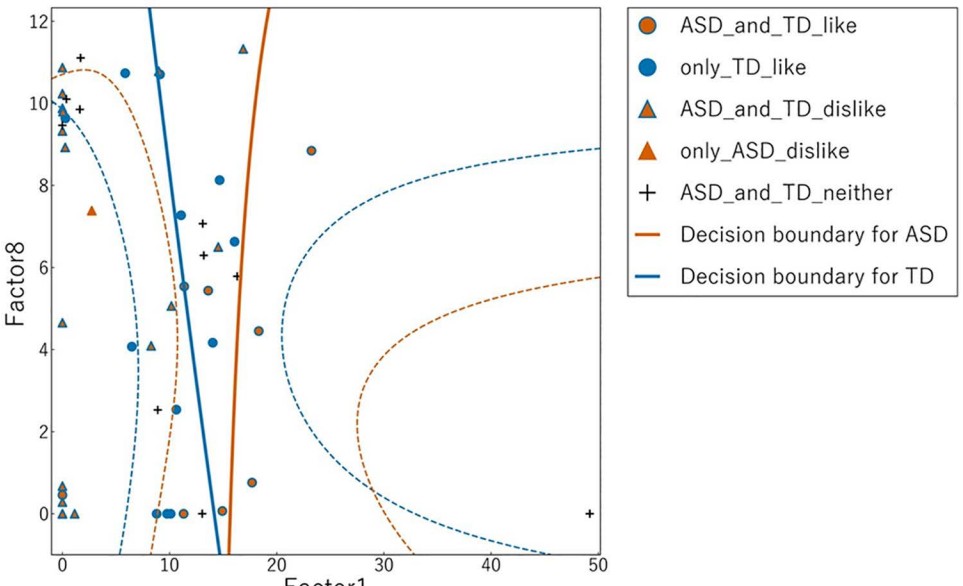

**Fig 5. Scatter plot of sunny-side-up egg images based on NMF Factor 1 and Factor 8.** The horizontal axis represents Factor 1 and the vertical axis represents Factor 8. Blue markers represent ratings by the ASD group, and orange markers represent ratings by the TD group. Circles represent images rated as "Liking," whereas triangles represent images rated as "Disliking." Solid lines show the linear regression boundaries between "Liking" and "Disliking" ratings for each group (blue for ASD and orange for TD). Dotted lines represent the 95% confidence intervals for these boundaries. ASD, autism spectrum disorder; TD, typical development; NMF, non-negative matrix factorization.

**Table 3. Mean values of the NMF factors for each preference category.**

| Label | Factor1 | Factor 6 | Factor 8 |
|---|---|---|---|
| ASD and TD groups like | 13.81 | 1.07 | 3.19 |
| only TD group like | 9.74 | 4.39 | 5.32 |
| ASD and TD groups dislike | 3.17 | 8.98 | 5.39 |
| only ASD group dislike | 2.74 | 10.00 | 7.38 |
| ASD and TD groups neither like nor dislike | 11.75 | 5.73 | 6.22 |

NMF, non-negative matrix factorization; ASD, autism spectrum disorder; TD, typical development.

noteworthy distinctions emerging between the ASD and TD groups. Results indicated the potential to predict preferences while elucidating the causes of selective eating behaviors, thereby offering solutions for individuals with ASD.

Quantitative visual preference ratings revealed specific differences in food image preferences between individuals with ASD and those with TD. While both groups exhibited analogous dislikes, individuals with ASD displayed a more restricted range of preferences for images they liked compared to TD individuals. The results of linear regression analysis suggest that the discriminant boundary for individuals with ASD is consistently positioned to the right of that for TD individuals in both Factor 1 and Factor 6 regression analysis, as well as in Factor 1 and Factor 8 regression analysis, indicating that individuals with ASD possess a narrower range of preferences. These findings are consistent with previous studies highlighting that individuals with ASD exhibit food selectivity or restricted food repertoires [32,33].

Both the questionnaire responses and the results from liner regression analysis indicate the importance of Factor 1 in sunny-side-up egg preference among individuals with ASD, supporting the tendency for such individuals to favor typical

images (i.e., clean with double-circle shapes and no burn marks). This aligns with previous research [34] that noted a preference for typical images among individuals with ASD.

Responses to artificial stimuli yielded additional insights into visual preferences for food. Despite the aim to represent idealized sunny-side-up eggs, these stimuli received varying evaluations. Notably, the CG image (Fig 2, third row from the top, sixth from the left), despite its meticulously controlled shape and surface, was not favored by either group. This observation suggests that digital perfection alone may not suffice for food acceptance, with other visual attributes captured through NMF analysis (represented by Factors 1, 6, and 8) proving more critical. The differential responses to food samples between the ASD and TD groups further corroborate the main finding that individuals with ASD generally demonstrate more restricted preferences. These findings carry practical implications for food presentation, indicating that artificial perfection may be less appealing than naturally occurring visual features that align with prototypical expectations.

This research suggests that utilizing NMF and decision tree analysis can elucidate the reasons and influences underlying food preferences among individuals with ASD. Importantly, the findings from linear regression analysis reveal dietary implications accessible to individuals with ASD from a visual perspective. Individuals may avoid specific foods or limit their food intake due to apprehensions regarding negative outcomes such as choking, vomiting, or stomach discomfort—a behavior frequently rooted in prior traumatic food experiences. By evaluating the visual characteristics of food in advance, calculating their significance, and establishing discriminant boundaries, it may be feasible to predict food preferences while also averting traumatic encounters.

The study is subject to certain limitations that warrant consideration in future research. First, the sample size was relatively small. Future studies with larger sample sizes are required to provide more robust and generalizable data on the potential use of statistical methods to investigate disparities and factors underlying visual preferences. Second, the examination was confined to one food item (i.e., sunny-side-up eggs). Although the choice of sunny-side-up eggs was predicated on their globally recognized simple shape, future research should incorporate a broader range of food items to evaluate the effectiveness of the employed methodology. Third, the analysis concentrated solely on visual aspects, whereas food preferences are acknowledged to be influenced by multiple sensory factors, including texture, smell, and taste. Finally, although both real and artificial stimuli were included, the limited number of artificial samples (two food samples and one CG image) constrains conclusions regarding the impact of artificial food representations.

## Conclusions

This study provides novel insights into the visual determinants of food preferences in individuals with ASD through systematic analysis of image features. Our findings reveal that individuals with ASD exhibit more restricted preferences for food images, with heightened sensitivity to prototypical visual characteristics. The identification of key visual features influencing food acceptance could facilitate caregivers and clinicians in developing more effective strategies for broadening food repertoires. These results bear considerable practical significance for addressing selective eating behaviors in ASD. Future investigations should extend this approach to a wider variety of foods and integrate additional sensory modalities to foster a more comprehensive understanding of food selectivity in ASD. Such investigations could ultimately aid in the formulation of evidence-based interventions for managing selective eating behaviors in individuals with ASD. As a practical application, we aim to provide caregivers of individuals with ASD with information on food preferences so that they can offer appropriate meals. We will also evaluate the effectiveness of this intervention in supporting healthier eating habits.

## Supporting information

**S1 Appendix. Cumulative contribution ratio of principal contribution analysis (PCA) for the list of interview questions.**
(DOCX)

**S2 Appendix.  Comparison of the explanatory power between Non-negative Matrix Factorization (NMF) and Principal Component Analysis (PCA).**
(DOCX)

**S1 Table.  Contributions of each principal component.**
(DOCX)

**S2 Table.  Adjusted R2 for regression analysis.**
(DOCX)

**S1 Data.  Participant demographics and response data.**
(XLSX)

**S1 Checklist.  Human Participants Research Checklist.**
(DOCX)

**S1 File.  A copy of the original ethics Original.**
(PDF)

**S2 File.  A copy of the original ethics English.**
(PDF)

**S3 File.  A copy of questionnaire Original.**
(PDF)

**S4 File.  A copy of questionnaire English.**
(PDF)

**S5 File.  All data usnderlying the findings described in their manuscript.**
(DOCX)

## Acknowledgments

We sincerely thank the participants and their families who participated in this study.

## Author contributions

**Conceptualization:** Kazunori Terada, Hirokazu Kumazaki.

**Data curation:** Kazunori Terada.

**Formal analysis:** Kazunori Terada.

**Methodology:** Kazunori Terada, Hirokazu Kumazaki.

**Project administration:** Hirokazu Kumazaki.

**Software:** Kazunori Terada.

**Supervision:** Kazunori Terada.

**Validation:** Kazunori Terada, Kazuhiro Ueda, Hirokazu Kumazaki.

**Visualization:** Kazunori Terada.

**Writing – original draft:** Kazunori Terada, Taku Imaizumi, Kazuhiro Ueda, Natsuki Nishikawa, Haruto Yoshida, Yukiya Taki, Shunsuke Fujii, Lu Li, Masashi Komori, Kunihito Kato, Hirokazu Kumazaki.

**Writing – review & editing:** Kazunori Terada, Kazuhiro Ueda, Hirokazu Kumazaki.

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
