## [Decision Letter · Decision Letter 0]

6 Apr 2025

Dear Dr. Kumazaki,

Thank you for submitting your manuscript to PLOS ONE. After careful consideration, we feel that it has merit but does not fully meet PLOS ONE’s publication criteria as it currently stands. Therefore, we invite you to submit a revised version of the manuscript that addresses the points raised during the review process.

We look forward to receiving your revised manuscript.

Kind regards,

Claudia Brogna

Academic Editor

PLOS ONE

 [This work was partially supported by a Grants-in-Aid for Scientific Research from the Japan Society for the Promotion of Science (23H04358).]. 

4. In the online submission form, you indicated that [Data are available from corresponding author upon reasonable request.].

Reviewers' comments:

Reviewer's Responses to Questions

**Comments to the Author**

1. Is the manuscript technically sound, and do the data support the conclusions?

Reviewer #1: Yes

Reviewer #2: Yes

2. Has the statistical analysis been performed appropriately and rigorously?

Reviewer #1: Yes

Reviewer #2: Yes

3. Have the authors made all data underlying the findings in their manuscript fully available?

Reviewer #1: Yes

Reviewer #2: Yes

4. Is the manuscript presented in an intelligible fashion and written in standard English?

Reviewer #1: Yes

Reviewer #2: Yes

Reviewer #1: It’s my pleasure to review the manuscript

Good work.

Well written manuscript, clear abstract, good discussion and good work

Aim is well explained and Title is good, grammatically is good no must

Language good

Reviewer #2: The authors conducted a Visual Feature Analysis (VFA study of how individuals (18-30 yrs) com Autism Spectrum Disorder (ASD) and Typical Development (TD) perceive and respond to visual stimuli, using a test based on 50 images of sunny-side-up eggs. In the context of Selective Appetite in Individuals with Autism Spectrum Disorder (ASD). The manuscript is clear and well-written; the authors thoroughly describe all study steps, including subject selection, group criteria, image development, test procedures, and statistical analysis.

The images, figures, and tables are well-presented.

My only concern are:

- How number of subjects were defined?

- Although the test does not have a direct application in practice, how do the authors suggest that it could evolve into a practical application?

**Do you want your identity to be public for this peer review?** For information about this choice, including consent withdrawal, please see our Privacy Policy

Reviewer #1: **Yes: ** I accept the manuscript for publication

Reviewer #2: No

---

## [Author Response · Author response to Decision Letter 1]

15 Apr 2025

Reviewer #1: It’s my pleasure to review the manuscript

Good work.

Well written manuscript, clear abstract, good discussion and good work

Aim is well explained and Title is good, grammatically is good no must

Language good

Thank you for commending our study.

Reviewer #2: The authors conducted a Visual Feature Analysis (VFA study of how individuals (18-30 yrs) com Autism Spectrum Disorder (ASD) and Typical Development (TD) perceive and respond to visual stimuli, using a test based on 50 images of sunny-side-up eggs. In the context of Selective Appetite in Individuals with Autism Spectrum Disorder (ASD). The manuscript is clear and well-written; the authors thoroughly describe all study steps, including subject selection, groupcriteria, image development, test procedures, and statistical analysis.

The images, figures, and tables are well-presented.

Thank you for commending our study.

My only concern are:

- How number of subjects were defined?

Thank you for pointing this out. In this study, we conducted an a priori power analysis using G*Power within a linear regression framework, assuming a medium effect size (f² = 0.15), an alpha level of .05, and a power of .80. This analysis indicated that approximately 77 participants (i.e., about 39 participants per group) would be required. Considering potential missing data or insincere responses into account, we aimed to recruit 50 participants per group. Therefore, we have added the two following texts in Participants part of Materials and methods section.

“We conducted an a priori power analysis using G*Power within a linear regression framework, assuming a medium effect size (f² = 0.15), an alpha level of .05, and a power of .80. This analysis indicated that approximately 77 participants (i.e., about 39 participants per group) would be required. Considering potential missing data or insincere response into account, we aimed to recruit 50 participants per group.”

- Although the test does not have a direct application in practice, how do the authors suggest that it could evolve into a practical application?

As a practical application, we aim to provide caregivers of individuals with ASD with information on food preferences so that they can offer appropriate meals. We will also evaluate the effectiveness of this intervention in supporting healthier eating habits. Therefore, we have added the following text in the Conclusion section.

“As a practical application, we aim to provide caregivers of individuals with ASD with information on food preferences so that they can offer appropriate meals. We will also evaluate the effectiveness of this intervention in supporting healthier eating habits.”

---

## [Editor Report · Decision Letter 1]

14 May 2025

Visual Feature Analysis on Selective Appetite in Individuals with Autism Spectrum Disorders

PONE-D-24-60413R1

Dear Dr. Hirokazu Kumazaki 

<table border="0" cellpadding="0" cellspacing="0" class="datatable3" style="border-collapse: collapse; width: 678.4px; line-height: 14px; color: rgb(0, 0, 51); font-family: verdana, geneva, arial, helvetica, sans-serif; font-size: 11.2px;"> <tbody> <tr style="background-color: rgb(244, 244, 244);"> <td style="padding: 3px; border: 1px solid rgb(255, 255, 255); width: 196.475px;"> </td> <td style="padding: 3px; border: 1px solid rgb(255, 255, 255);"> </td> </tr> </tbody></table>

We’re pleased to inform you that your manuscript has been judged scientifically suitable for publication and will be formally accepted for publication once it meets all outstanding technical requirements.

Kind regards,

Claudia Brogna

Academic Editor

PLOS ONE

---

## [Editor Report · Acceptance letter]

PONE-D-24-60413R1

PLOS ONE

Dear Dr. Kumazaki,

I'm pleased to inform you that your manuscript has been deemed suitable for publication in PLOS ONE. Congratulations! Your manuscript is now being handed over to our production team.

Kind regards,

on behalf of

Dr. Claudia Brogna

Academic Editor

PLOS ONE